# Mental Strategies in a P300-BCI: Visuomotor Transformation Is an Option

**DOI:** 10.3390/diagnostics12112607

**Published:** 2022-10-27

**Authors:** Nikolay Syrov, Lev Yakovlev, Varvara Nikolaeva, Alexander Kaplan, Mikhail Lebedev

**Affiliations:** 1V. Zelman Center for Neurobiology and Brain Restoration, Skolkovo Institute of Science and Technology, 121205 Moscow, Russia; 2Baltic Center for Neurotechnology and Artificial Intellect, Immanuel Kant Baltic Federal University, 236016 Kaliningrad, Russia; 3Laboratory for Neurophysiology and Neuro-Computer Interfaces, Human and Animal Physiology Department, School of Biology, M.V. Lomonosov Moscow State University, 119234 Moscow, Russia

**Keywords:** brain-computer interface, visuomotor transformation, motor imagery, event related potentials, neurorehabilitation

## Abstract

Currently, P300-BCIs are mostly used for spelling tasks, where the number of commands is equal to the number of stimuli that evoke event-related potentials (ERPs). Increasing this number slows down the BCI operation because each stimulus has to be presented several times for better classification. Furthermore, P300 spellers typically do not utilize potentially useful imagery-based approaches, such as the motor imagery successfully practiced in motor rehabilitation. Here, we tested a P300-BCI with a motor-imagery component. In this BCI, the number of commands was increased by adding mental strategies instead of increasing the number of targets. Our BCI had only two stimuli and four commands. The subjects either counted target appearances mentally or imagined hand movements toward the targets. In this design, the motor-imagery paradigm enacted a visuomotor transformation known to engage cortical and subcortical networks participating in motor control. The operation of these networks suffers in neurological conditions such as stroke, so we view this BCI as a potential tool for the rehabilitation of patients. As an initial step toward the development of this clinical method, sixteen healthy participants were tested. Consistent with our expectation that mental strategies would result in distinct EEG activities, ERPs were different depending on whether subjects counted stimuli or imagined movements. These differences were especially clear in the late ERP components localized in the frontal and centro-parietal regions. We conclude that (1) the P300 paradigm is suitable for enacting visuomotor transformations and (2) P300-based BCIs with multiple mental strategies could be used in applications where the number of possible outputs needs to be increased while keeping the number of targets constant. As such, our approach adds to both the development of versatile BCIs and clinical approaches to rehabilitation.

## 1. Introduction

Cerebral stroke is the leading cause of human disability globally. Notwithstanding the progress made in rehabilitation methods, up to 30% of patients remain immobilized [1,2]. Therefore, further research is needed on methods that could contribute to brain plasticity in post-stroke patients. One of the main drawbacks of existing neurorehabilitation techniques is an insufficient utilization of neural feedback associated with activity in the affected and intact brain areas. Brain–computer interfaces (BCIs) offer a potentially powerful strategy for rehabilitation because they offer a variety of neural feedbacks. A therapeutic BCI translates the mental efforts of a patient into commands to external assistive devices to improve physiological functions affected by stroke [3]. It has been shown that rehabilitation is more effective during such BCI operations compared to traditional physical therapy [4,5,6].

Here, we propose that P300-BCIs be added to the existing rehabilitation approaches. P300-BCIs are based on the detection of changes in stimulus-locked EEG potentials (called event-related potentials or ERPs). Such BCIs are effectively used as assistive technologies for patients with motor impairments but not for sensorimotor rehabilitation per se. In clinical approaches, the BCIs that are used that are mostly based on the detection of desynchronization in sensorimotor EEG rhythms during motor imagery (MI) [1]. MI activates neural circuits engaged in motor control, which leads to plastic modifications of these circuits [1]. Yet, the information transfer rate is low in motor-imagery BCIs, which limits their practical significance. By contrast, the P300 paradigm offers better information transfer rates, accuracy and reliability. To increase the number of output bits in a P300-BCI, one has to increase the number of external stimuli. A drawback of having too many external stimuli is that BCI operations slow down because more time is needed to collect the number of samples required for good performance [7]. Overall, P300-based BCIs are much easier to operate in comparison to motor imagery-based BCIs [8,9], which makes them potentially very useful for patients whose neurological functions are affected by stroke. Yet, a clinical BCI should be designed in such a way that it is not only user-friendly but also affects sensorimotor neural circuits in an appropriate way.

During the operation of a P300-BCI, the user typically mentally counts the number of times the target stimulus flashes. This mental task assures that the subject attends to the target and engages basic arithmetic and language abilities. However, simply counting may not be adequate for rehabilitation purposes because this mental strategy does not necessarily address the specific disabilities of a patient—particularly the disabilities related to motor functions of the limbs. Here, we enriched the traditional P300 paradigm with an additional mental task that specifically engaged the sensorimotor circuitry. The task consisted of cued MI, where subjects started to imagine a movement following a stimulus onset. This task added a motor-related component to the BCI and also increased the number of BCI commands without increasing the number of external stimuli. With two targets and two mental reactions—counting and motor imagery—the BCI had four outputs. We hypothesized that ERPs induced during cued MI would significantly differ from the counting-related ERPs, which is consistent with previous work [10,11,12]. For the cued MI, we expected an activation of both visual and sensorimotor areas through the process of visuomotor transformation (also known as stimulus-response mapping). Additionally, MI ability could be impaired in stroke patients [13,14], so visually induced motor imagery could be easier to perform compared to traditional tasks. Indeed, external factors such as visual cues can be used to trigger and visually guide motor imagery in patients [15]. The cued MI that we implemented here has been shown to engage cortical motor areas in a very similar way to real movements [16,17]—that is, thinking about movements without executing them is good enough for BCI and/or rehabilitation purposes, particularly in patients who cannot move. In paraplegics, attempts to move increase activity in the primary motor cortex, parietal lobe and cerebellum, as shown using fMRI [18]. Attempted movements cause EEG responses in patients as well, and these responses are stronger compared to kinesthetic MI [19,20]. A BCI was previously reported in which motor attempts were decoded from continuous EEG recordings in six severely affected chronic stroke patients [21]. Finally, a recent meta-analysis showed that BCIs that use movement attempts work better for motor rehabilitation compared to ones that rely on MI [22]. Based on these considerations, the aim of the present study was to assess the operation of a two-stimuli P300-BCI that incorporated both motor and non-motor mental strategies. This hybrid paradigm offers advantages for post-stroke rehabilitation practice because it works fast and reliably and can target different brain networks while using a limited number of stimuli. We hypothesized that a sensorimotor potential (SP) would likely occur during cued MI but not during counting [23,24]. To enhance the discrimination ability within the proposed BCI paradigm, we used common spatiotemporal patterns (CSTPs) as a feature extraction method. This algorithm is similar to the common spatial pattern (CSP) method that separates different conditions using spatial filters [25]. CSP is suitable for BCIs that use traditional MI [26]. However, CSP is time-invariant as it does not consider the temporal characteristics of ERPs. CSTP is more suitable for the decoding of transient EEG patterns because it considers both spatial and temporal features [27,28,29,30].

## 2. Materials and Methods

### 2.1. Participants

Sixteen healthy volunteers participated in the study (mean age 24 ± 3 years; 4 females and 12 males). All of them were informed of their rights and gave informed consent for participation in the study. The experimental procedures were approved by the Lomonosov Moscow State University Committee for bioethics (protocol no. 111-ch, the approval date: 19 June 2020). The study followed the Declaration of Helsinki Ethical Principles for Medical Research Involving Human Subjects. All the subjects were informed about the study procedures and gave their written consent to participate. 

### 2.2. Study Design, Procedures and Tasks

The subjects sat in a chair in a comfortable position with their hands placed on a platform that had two buttons (5 cm in diameter and 10 cm in height) installed (Figure 1). The buttons were used for the delivery of visual cues and as a target for the performance of mental presses. The left and right buttons corresponded to the right and left hand, respectively. During experimental runs, the button lights were turned on in a pseudorandom order to produce the external visual stimuli for the BCI. 

The experimental sessions consisted of four runs—that is, two runs for each of the two experimental conditions: mental counting (MC) and motor imagery (MI). The order of runs was selected at random for each subject. For MC, participants were instructed to count how many times the target button was lit up. For MI, they were asked to imagine pushing the button with their hand in response to the visual cue. Each run was constructed of 6 trials. The trials started with the word "left" or "right" appearing on the 22-inch LCD monitor placed in the front of the subject. This instruction lasted for 5 s and was generated randomly for each trial. Next, the buttons were lit up in random order while the subject attended to the instructed target. The stimuli were presented 30 times; half of the stimuli corresponded to the target. Stimulus duration was 200 ms and the interstimulus interval was 800 ms. The experimental sequence is clarified in Figure 1B.

### 2.3. Signal Acquisition and Processing

EEGs were recorded using an NVX52 DC amplifier (MKS, Zelenograd, Russia) with 22 Ag/AgCl electrodes mounted over the following sites: Fp1, Fp2, Fz, FC3, FCz, FC4, C5, C3, Cz, C4, C6, CP3, CPz, CP4, P3, Pz, P4, PO3, POz, PO4, O1 and O2. The average of channels A1 and A2 was used as the reference. The electrode–skin impedance did not exceed 20 kΩ. The signal was sampled at 1000 Hz. During the preprocessing procedure, the EEG signals were filtered using a bandpass filter with a frequency range of 1–15 Hz and then downsampled to 250 Hz. 

### 2.4. Feature Extraction

#### 2.4.1. Raw Epochs Concatenation

The filtered signal was segmented into 1-second epochs (0–1 s after the stimulus onset). By choosing epochs of this length, we included in the analysis both early visual components and late cognitive potentials related to the performance of the mental tasks (MC or MI). Once the EEG was segmented, the epochs from all channels were stacked into one vector of features. For each subject, 360 target-stimulus epochs were collected (180 for MC and 180 for MI) and 360 non-target-stimulus epochs. Signal preprocessing and analysis were conducted using custom Python scripts that were based on the open-source EEG analysis package MNE v1.1 [31].

#### 2.4.2. Common Spatio-Temporal Filtering

The pipeline of combination for the spatio-temporal features was adopted from [30]. The common spatial patterns (CSP) method was commonly used for the extraction of motor imagery-related features from multi-channel EEGs in MI-based BCI [25,32,33]. CSP implies the linear transformation of the multi-channel EEG data of two different conditions into low-dimensional common spatial subspace by a set of the spatial filters found in the way to maximize the variance of two-class signal matrices. Overall, the CSP algorithm projects X to spatially filtered Z as:Z = XW(1)
where the rows of the projection matrix W, of which each row is the spatial filter, consists of weights for channels.

To create sensitive spatial filters, we preprocessed the signal of each subject by band-pass filtering in the 1–15 Hz range; then, the signal was downsampled to 250 Hz and segmented into 1 s epochs (0–1 s after the stimulus onset). While CSP implies the separation of a pair of classes, we performed CSP three times for the following pairs: MC vs non-target, MI vs non-target and MI vs MC. A further, detailed description of the spatio-temporal filtering algorithm uses an example of a pair of conditions MC and MI; also, the filtering pipeline is pictured on Figure 2. For each pair, the following steps were performed:

First, we calculated a pair of covariance matrices of size N × N, where N denotes the number of channels [34]:R_MI_ = X_MI_ X_MI_^T^ trace (X_MI_ X_MI_^T^); R_MC_ = X_MC_ X_MC_^T^ trace (X_MC_ X_MC_^T^)(2)
where X_MI_ and X_MC_ denote the preprocessed multichannel EEG epochs under motor imagery and mental count conditions with the dimensions N × T—where T is the number of samples per channel; trace(X) gives the sum of the diagonal elements of X. To obtain robust filters, we cleaned the covariance matrices by removing unrepresentative «noisy» epochs by the procedure suggested in [35].

Then, we built a composite covariance matrix that could be factored by GED (generalized eigendecomposition) as:R = R_MI_ + R_MC_ = U_0_ ΣU_0_^T^(3)
where U_0_ is the matrix of eigenvectors and Σ is the diagonal matrix of the corresponding eigenvalues ranging from higher to lower.

Then, we used whitening to transform the average covariance matrices as:RW_MI_ = PR_MI_ P^T^; RW_MC_ = PR_MC_P^T^(4)
where P is the whitening matrix, obtained as:P = Σ^−1/2^ U_0_^T^(5)

RW_MI_ and RW_MC_ share common principal components (eigenvectors) and the sum of corresponding eigenvalues for the two matrices will be equal to one [36]. Thus, they can then be factorized as:RW_MI_ = UΣ_MI_ U^T^; RW_MC_ = UΣ_MC_ U^T^(6)
where U is the matrix of the common principal components (eigenvectors) and denotes the diagonal matrix of eigenvalues. The eigenvalues are sorted into descending order and the eigenvectors with the largest eigenvalues for RW_MI_ have the smallest eigenvalues for RW_MC_ and vice versa.

The CSP projection matrix is defined as:W = U^T^ P(7)
which transforms the original EEG data from the used pair of epochs sets (X) into uncorrelated components (Z, see Equation (1)).

Z can be interpreted as the source components of a complex mixture of signals from the original EEG data, including common and specific components of different mental states (state A and state B). The columns of W-1 (pseudo-inverse W) are spatial patterns that can be interpreted as distribution vectors of the sources of EEG variations [37], i.e., topographic forward projections to the scalp of the principal components. The first and last columns of W-1 are the most important spatial patterns that explain the largest variance of one state and the smallest variance of the other [38]. In such a case, to increase the signal-to-noise ratio, only eight spatial filters—i.e., columns of W, corresponding to the 4 largest and 4 smallest eigenvalues—were used for the data transformation (calculated by the dot product X and selected columns of W, see Equation (1)). Thus, we get a pair of spatially filtered EEG epochs for each condition from the used pair Z_MI_ and Z_MC_, which are matrices of the size E × N* × T—where E and T are the number of epochs and number of samples, respectively—and N* denotes the number of selected spatial filters (equal to 8 in our case).

These data were further used to capture temporal features by the common temporal pattern (CTP) method. CTP, proposed by [29], is the temporal counterpart of the common spatial patterns in which time covariances are considered instead of space covariances. Thus, at the first step for the pair of CSP-filtered data Z_MI_ and Z_MC_, we calculated a pair of covariance matrices of the size T × T, where T denotes the number of samples.

Since the number of time samples is larger than the number of spatial projections (250 samples at 250 Hz sampling rate), the time covariances are not full-rank and thus not positive definite. Therefore, we downsampled the data to 100 Hz and then used regularization to make the covariance matrices full-rank. We used a covariance estimator with a shrinkage algorithm from the Python-MNE 0.23.0 module, with a constant regularization coefficient equal to 0.1.

The CTP method was used twice; one set of temporal filters was trained using the EEG epochs filtered through the first four CSP spatial filters (corresponding to the MI condition) and another set of CTP filters was trained using the EEG epochs filtered through the last four CSP filters, corresponding to the MC. Eight CTP filters (corresponding to the 4 largest and smallest eigenvalues) were taken from each of the two sets. All 16 filters were applied to CSP-processed epochs.

The above procedure was done in each pair of conditions separately. Overall, 48 CTP filters were obtained. Finally, a Random-Forest classifier was trained. During the cross-validation procedure, both the CSP and CTP filters were calculated using the training dataset and were applied to the test epochs.

### 2.5. Offline Classification

The BCI accuracy was estimated offline using an algorithm that simulated P300-BCI control by selecting the target based on several presentations of the target and non-target stimuli. In this algorithm, 3 parts of each subject’s dataset were analyzed: 90 samples for the targets from the mental counting task, 90 samples for the targets from the cued MI task and 90 samples for non-targets (with an equal ratio for all conditions). To generate a prediction, we used 4 non-target epochs and 4 target epochs (MC or MI) as test data, and the remaining samples were used as training data for the decoding algorithm. The training data were used to construct the CSP and CTP filters and to fit the prediction model. 

The prediction was defined as correct if two conditions were met:In the 4 target presentations, the number of correct classifications was greater than the number of errors;In the 4 non-target presentations, the number of their correct classifications as non-targets was greater than the number of errors, or the number of errors for non-targets was less than the number of correctly classified targets.

Accuracy scores were computed for each subject via 20 repetitions of the prediction. We used the “one vs one” classification approach with a random forest (RF) classifier (the number of models was equal to 100). The implementation of the classifier was taken from SciKit-Learn Python library v0.23.2 [39].

## 3. Results

To assess cortical responses to different stimuli and instructions, we started by computing the across-subject average ERPs for each mental strategy. Figure 3A shows the average ERPs for the target (left) and non-target stimuli (right), separately for the MI and MC strategies (color-coded traces). The strategy-dependent differences in ERP amplitude and shape were considerable for the target responses and subtle for the non-target responses. 

As evident from Figure 3A, strategy-dependent ERP modulations could be quantified as P2, N2 and the late positive component SP—spanning the 500–800 ms interval after the stimulus. Note that the SP component had the same characteristics (in terms of topography and time-amplitude characteristics) as the motion-related reafferent potential (also known as Go-P3 or motor-related potential) that occurs when a participant responds to a stimulus with an actual movement [23,40,41]. For these overt-action tasks, the SP component is the strongest over the hemisphere contralateral to active limb. Yet, in our experiment with cued MI, the strongest response was over the central midline area (Cz electrode, see Figure 3B).

To compare the two approaches to the feature extraction (raw-data vs CSTP), we conducted offline predictions; predictions were conducted 20 times for each approach. For that most of the participants (14), we achieved a classification accuracy above chance level for both the raw-data features (mean accuracy = 0.6 +/− 0.18 and Cohen’s kappa = 0.37) and the CSTP (mean accuracy = 0.76 +/− 0.17 and Cohen’s kappa = 0.51; see Figure 4A). The chance level p0 was set to 0.33, according to the assumption that all agreements occur by chance and was computed using the obtained confusion matrices [42,43]. The experimental chance level with the fully random classifier’s outputs was equal to 0.16. We also found that accuracy significantly increased when CSTP features were used (Wilcoxon signed-rank test, *p* = 0.018). 

Figure 4B presents the confusion matrices for the accuracy of prediction with the two approaches. It can be seen that the different mental strategies (MC versus MI) were distinguished by the classifiers with good accuracy. At the same time, the input vectors of raw features related to the non-target epochs were often misclassified as targets and vice versa. False positive and false negative rates were almost the same for the MC and MI conditions. By contrast, the use of CSTP features lowered the number of false negative cases.

To test the hypothesis that the discrimination ability of the proposed paradigm depends on the presence of SP during the cued motor imagery, we examined in greater detail the contribution of different ERP components to decoding. First, we ran predictions using the raw signal from only one EEG electrode. We then repeated the accuracy estimation procedures for all the electrodes to evaluate the contribution of data from each site. Next, we visualized the mean accuracy as a topographic map (Figure 5). The Friedman test showed that the accuracy scores were significantly different across the electrodes (χ^2^ = 66.1; *p* < 0.001). The post-hoc Wilcoxon tests (Bonferroni-corrected *p* = 0.002) showed that the accuracy scores achieved with data from the occipital sites and the channels Cz and C6 were significantly higher than the chance level.

The electrodes with the maximal classification accuracy were concentrated over the occipital–parietal (O2, PO3, PO4 and P4) and central (Cz and C6) scalp areas. Notably, the accuracy provided by single-channel data could be as high as 0.6.

The CSTP-features used in the classification included spatial patterns obtained by applying temporal filters to the CSP-processed epochs. Temporal filters are matrices with eigenvectors of size T × T (101 × 101 in our case). Only the first and last four columns were used in this study. Temporal filters transform the CSP-preprocessed time-series epochs into a number of components by linear combinations with corresponding filter weights. The output components were in the form of spatial maps (matrices with a size equal to the number of EEG channels), which are linear combinations of samples for each sensor [44]. Each component had a corresponding temporal pattern obtained by inversion of the matrix with the temporal filters. Visualization of CSTP-filtered epochs can help us to understand which features contributed to the classification (see Figure 6).

It can be seen that the spatial maps produced by the CSTP transformation have a dipolar structure and reveal differences between the MI and MC conditions. The temporal patterns (inverted temporal filters) depicted in Figure 6B capture the temporal structure and highlight the temporal features; the two conditions are distinguished by maximizing the difference in signal variance. It can be seen that the maximal coefficients corresponded to the temporal features whose latency matched the distinct ERP components (P2, N2 and SP). In particular, the first MI-related temporal patterns captured the components that corresponded to N2 and SP.

## 4. Discussion

While research on ERP-based BCIs typically focuses on the designing of stimuli (visual, auditory and tactile) that match the requirements of a particular application and the development of high-performance algorithms for feature extraction and classification [45,46,47], less attention is paid to the mental tasks that participants use when operating this type of BCI. In particular, motor-related strategies are rarely used because ERP-based BCIs are mostly used for communication purposes, not for tasks that involve limb movements. Limb-movement BCIs typically involve MI, but not ERPs. In this study, we merged ERP and MI-based designs. We developed a hybrid BCI paradigm based on visual-evoked potentials, where we increased the number of possible outputs not by adding new stimuli, but by adding a mental response to the stimulus that required MI.

We investigated the ERPs that occurred when the participants were asked to alternate between two mental strategies: mentally counting the target stimuli and imagining hand movements directed toward the stimuli. Strategy changes were manifested in several ERP components with different latencies: P2, N2 and central-localized SP. The spatio-temporal characteristics of the P2 and N2 components that we observed were similar to the effects observed in the Go/NoGo paradigm [12]. Additionally, P2 (early visual potential, peaking at around 200 ms) is known to be sensitive to spatial attention [48]. In P300-BCIs, this ERP component is used to detect the difference between target and non-target ERPs, and occipital sites provide the best classification accuracy (see Figure 5). The component P2 can be also seen in the CSTP-patterns presented in Figure 6B. As for the frontal N2, its amplitude reflects the amount of cognitive control resources required to carry out the task [49]; this component contributes to the inhibition of motor responses [50]. In this study, we found that N2 was stronger during mental counting than during MI tasks; there was also a difference in the latencies. This component could be also found in the CSTP-patterns, which confirms that N2 contributed to the distinguishing between MI and MC conditions carried out by the classifier. SP was the component with the longest latency. The SP became stronger during MI— particularly over centrally located channels such as Cz (see Figure 5). Additionally, the MI-related CSTP pattern (see Figure 6B) had a component with a latency around 500–600 ms, which resembled a SP—particularly when the topography maps of the CSTP-processed epochs were considered (Figure 6A). This is a very promising finding; according to motor imagery and motor execution studies, SPs and SP-like potentials correspond to the activation of sensorimotor cortical areas [10,40]—so a cued MI-based BCI could be proposed for engaging these areas during the rehabilitation of post-stroke patients.

There is plentiful evidence in the literature of the possibility of modulating ERP components by mentally responding to external stimuli in different ways. For example, centrally localized late potentials change when participants are required to perform a mental rotation of observed human hands (visuo-motor imagery task) [51], and motor-related selective attention tasks—such as motor imagery and execution—improve classification accuracy in P300-based BCIs compared to numerical counting tasks [11]. Moreover, motor imagery enhances the reliability and accuracy of P300-based spellers [52], enhances the sense of agency during BCI control [53], and could be combined with the P300 paradigm for interactions with virtual objects in VR environments [54]. Thus, the merging of the P300 and MI paradigms opens new possibilities in practically relevant BCIs, including the development of P300-BCIs where the number of output commands is increased by enriching mental strategies instead of adding new external stimuli. In this study, four BCI commands were achieved with two stimuli that were presented four times each to achieve accurate classification with CSTP features (reached accuracy well enough for successful BCI control [55]). Reductions in the number of external stimuli decreases the user’s spatial attention load and reduces the time required to issue a BCI command. One benefit of such fast operations is that sensory feedback (in this case, visual) is not delayed. In this study, the stimuli presentation pace was relatively slow: 200 ms for stimulus presentation and 800 ms for the intervals between the stimuli. We chose this pace to measure ERP component development over a relatively long epoch and to avoid component overlay. Despite these arrangements, the classification accuracy was low for several participants. We suggest that several limitations affected the accuracy: First, the participants were naive in motor imagery, which resulted in small modulations of their ERPs when they attempted to imagine movements. Second, the long-lasting experimental runs could have caused fatigue, which in turn reduced MI performance. Third, in the paradigm described here, there were three classes to predict, whereas P300-based BCI typically distinguishes between just two: target and non-target—so the chance level was lower in our case.

Overall, the features of our paradigm make it useful for motor rehabilitation. Mental movement attempts performed in response to visual stimuli activate sensorimotor cortical areas through the mechanism of visuomotor transformation; the BCI implemented here makes use of features related to this mechanism. As such, this BCI could be used to restore visuomotor transformations in post-stroke patients. This development is novel, as ERP-based BCIs are rarely used as rehabilitative tools [56,57,58]; as such, the proposed approach still has to be tested in a clinic.

## 5. Conclusions

We demonstrated the possibility of increasing the number of commands in a classical P300-BCI based on the enriching of mental reactions to target stimuli by the requirement of imagining movements. Participants performed either mental counting of a target stimuli or motor imagery directed toward them. In future work, mental strategies could be enriched even further by endeavors such as quasi-movements, covert speech and others. The cued MI implemented here triggered visuomotor transformations. Visuomotor transformation engages multiple cortical areas that enact motor planning, motor preparation and control, and this type of engagement could be particularly useful for neurorehabilitation purposes.

## Figures and Tables

**Figure 1 diagnostics-12-02607-f001:**
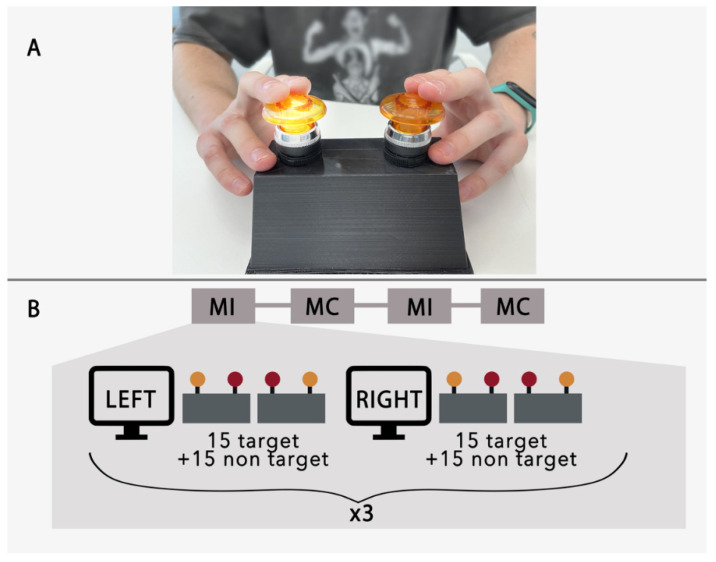
(**A**) Two press buttons with LEDs for button illumination. Participants held their hands on the buttons and responded to button highlights, when instructed, by imagining pressing the button. (**B**) Experimental sequence. “MI” and “MC” correspond to motor imagery and mental counting instructions.

**Figure 2 diagnostics-12-02607-f002:**
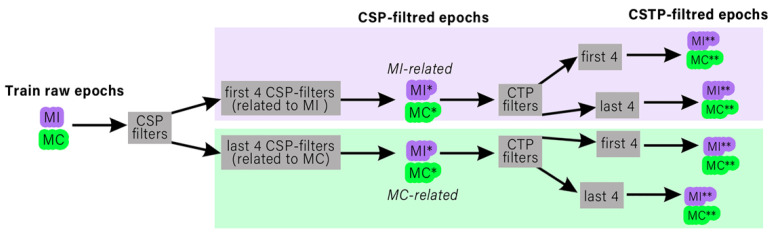
The pipeline of consequent spatial and temporal filtering within the feature extraction. The pair of sets of raw EEG epochs (MI and MC) were used to create CSP-filters; then, the raw epochs were processed by eight CSP filters (first four and last four). Two sets of spatially-filtered epochs (MI* and MC*) were used to create CTP filters; on the last step, eight CTP filters (corresponding to the 4 largest and smallest eigenvalues) were selected from each of the CTP filter matrices and the spatially processed EEG epochs were filtered through each one. MI** and MC** are the resulting CSTP-filtered epochs.

**Figure 3 diagnostics-12-02607-f003:**
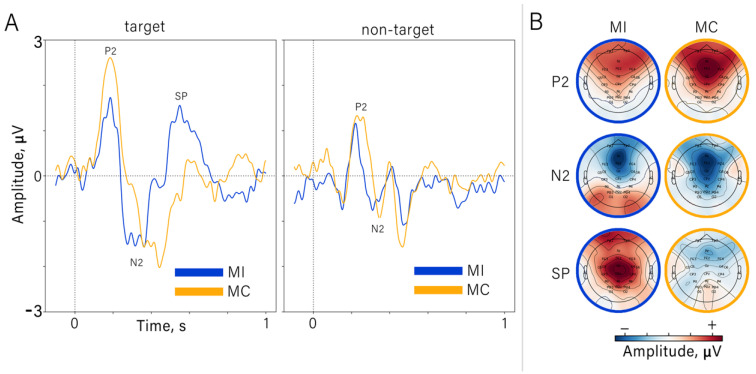
(**A**) ERPs for different stimulus types and mental strategies. P2, N2 and SP denote the consecutive ERP components. A: The across-subject average ERPs for the target (left) and non-target (right) stimuli for the two mental strategies: motor imagery (MI) and mental counting (MC; color-coded). Data from the electrode Cz are shown. (**B**) The topographic map of the ERP amplitude for the MI and MC conditions.

**Figure 4 diagnostics-12-02607-f004:**
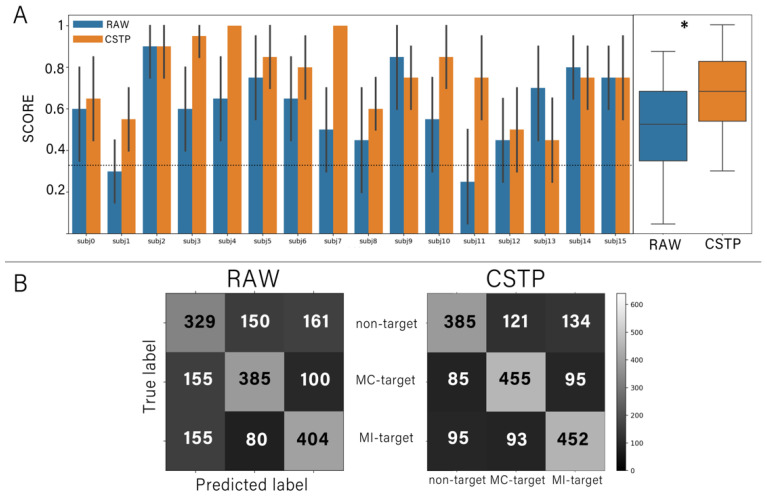
(**A**) Offline performance of each subject averaged over all cross-validation trials. The bars denote the mean; the error strokes denote the standard deviation. Asterisk (*) indicates significant difference in accuracy score between two feature extraction approaches (*p* = 0.018). The dashed line indicates the chance level. (**B**) Confusion matrices for each feature extraction approach, where the grayscale intensity and numbers denote the quantity of true positive, false positive, false negative and true negative classification cases.

**Figure 5 diagnostics-12-02607-f005:**
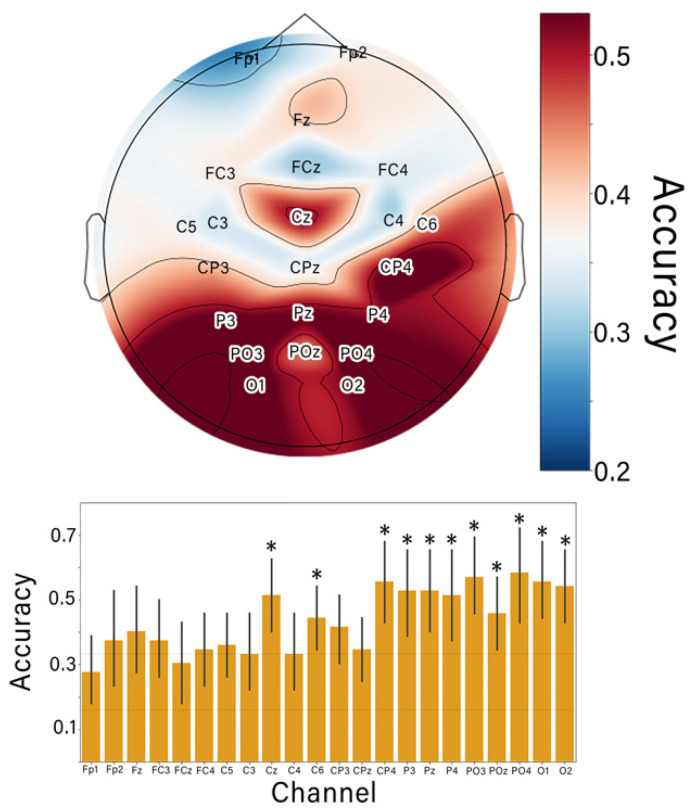
Scalp distribution of the average single-channel classification accuracy and accuracy values (an average for all participants and cross-validation trials). The colors represent the mean accuracy for single-channel classifications conducted offline, with red corresponding to high scores. The bars define the mean values of the offline estimated single-channel accuracy; the error bars show 95% confidence intervals. Asterisks indicate significant differences from the chance level 0.33 (Wilcoxon sign rank test with Bonferroni correction).

**Figure 6 diagnostics-12-02607-f006:**
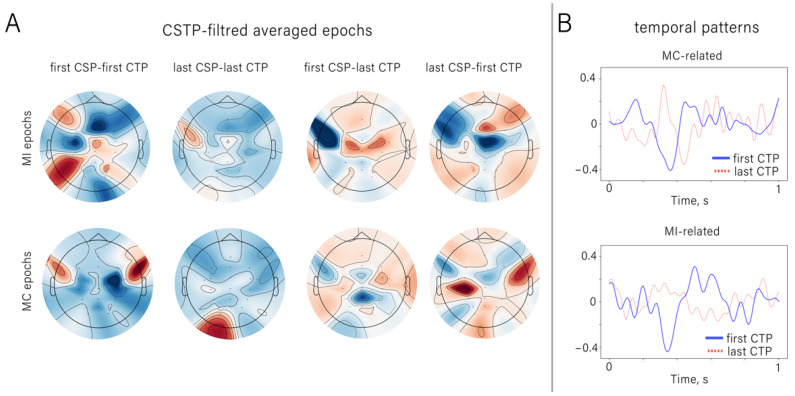
(**A**). Spatial patterns (spatial maps) of the CSTP-processed MI and MC epochs (median over all subjects, epochs and all cross-validation trials). The CSTP filters were obtained for the “MI vs MC” pairs. The captions above the columns of panels denote which spatial and temporal filters were used to obtain a particular component. For example, “first CSP—first CTP’’ means that this map resulted from applying the first four temporal filters from Wtemporal, which were obtained from the data processed by the first four spatial filters from Wspatial—which in turn are related to the MI condition, and vice versa. “Last CSP—last CTP” corresponds to the last four temporal filters from Wtemporal obtained from the data processed with the four MC-related spatial filters. (**B**). CSTP patterns of the first and last CTP filters (first and last four columns of Wtemporal (which correspond to the largest and smallest eigenvalues)) obtained from the data spatially filtered with the MI-related CSP filters (first columns of Wspatial) and MC-related CSP filters (last columns of Wspatial).

## Data Availability

The raw data supporting the conclusions of this study are available upon request from the corresponding author.

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
