# Peer review of "Mental Strategies in a P300-BCI: Visuomotor Transformation Is an Option"

_diagnostics, 2022, doi:10.3390/diagnostics12112607_

Round 1

Reviewer 1 Report

It is a very interesting paper that addresses a real and concrete problem: the need to reduce the large number of stimuli necessary to obtain a good classification in the P30O-based BCIs. For this, the authors propose the hybridization of the P300 and motor imagery paradigms.

Motor imagery produces changes in the sensorimotor rhythms of the EEG: mu (8-12Hz) and beta (18-30Hz). The reasons why the authors filter the signal between 1-15Hz are not clear. It is then necessary to justify why the beta band is not being used.

The incorporation of a graphic with the registration protocol, would help the reader in the understanding of section 2.2.

How was the chance level calculated? I suggest recalculating it taking into account the following bibliographical reference: Billinger, M., Daly, I., Kaiser, V., Jin, J., Allison, B.Z., Gernot, R.M., Brunner, C. 2012. Is It Significant ? Guidelines for Reporting BCI Performance. En: Allison, B. Z., Nijholt, A., Dunne, S., Leeb, R., Millán, J. D. R. (eds.), Towards Practical Brain-Computer Interfaces, pp. 333-354. Springer

Please, label the potentials of the Figure 3.A "non-target"

In figure 5, it would be convenient to show the accuracy values of each channel.

It is suggested that the figures be shown after they are introduced in the text.

In line 362 the explanation of the acronym SP is repeated. Also in line 273. The acronym was presented in line 97.

Please correct and unify the format for citing references throughout the article; for example line 85 page 1; lines 384 and 387 in page 11; etc.

Check the format of all references, for example 25, 38, 40 etc.

Author Response

Please find enclosed our revised manuscript, "Mental strategies in a P300-BCI: visuomotor transformation is an option".

We appreciate the time that you dedicated to providing the insightful and highly helpful comments regarding our manuscript. We have substantially changed the manuscript in following the reviewers' suggestions.

Please, find our responses to the particular comments . We hope that our manuscript will be found suitable for publication.

RW - Reviewer / AU - authors

[RW]: It is a very interesting paper that addresses a real and concrete problem: the need to reduce the large number of stimuli necessary to obtain a good classification in the P30O-based BCIs. For this, the authors propose the hybridization of the P300 and motor imagery paradigms.

[AU]: We thank the Reviewer for the positive comments regarding our manuscript. We hope that the edits we have made in accordance with the Reviewer’s suggestions will improve the manuscript further. 

[RW]: Motor imagery produces changes in the sensorimotor rhythms of the EEG: mu (8-12Hz) and beta (18-30Hz). The reasons why the authors filter the signal between 1-15Hz are not clear. It is then necessary to justify why the beta band is not being used.

[AU]: Thank you for the reasonable question. Indeed, the use of the beta-range could improve the BCI performance. However, in this particular manuscript we focused on the P300 design.  Accordingly, we used ERPs as features. In the case of ERPs, the beta band increases noise when it interferes with the ERPs. Additionally, the higher frequency EEG bands could be contaminated by muscle activity [1]. 

[RW]: The incorporation of a graphic with the registration protocol, would help the reader in the understanding of section 2.2.

[AU]: Thank you for pointing this out. The schematics of the experimental pipeline has been added to Figure 1.

[RW]: How was the chance level calculated? I suggest recalculating it taking into account the following bibliographical reference: Billinger, M., Daly, I., Kaiser, V., Jin, J., Allison, B.Z., Gernot, R.M., Brunner, C. 2012. Is It Significant ? Guidelines for Reporting BCI Performance. En: Allison, B. Z., Nijholt, A., Dunne, S., Leeb, R., Millán, J. D. R. (eds.), Towards Practical Brain-Computer Interfaces, pp. 333-354. Springer

         [AU]: We thank the Reviewer for this suggestion and the very helpful reference. We have incorporated this suggestion. We have also explained that the chance level was computed from the confusion matrix. Additionally we estimated the empirical chance level that was obtained with randomly generated classifier outputs.

[RW]: Please, label the potentials of the Figure 3.A "non-target"

[AU]: Thank you for pointing this out. We have added these labels to  figure 3 A

[RW]: In figure 5, it would be convenient to show the accuracy values of each channel.

[AU]:   Thank you for this remark. The figure has been corrected according to this recommendation. We have recalculated the accuracy scores to obtain the full dataset for each channel. With this analysis, the score topographical distribution slightly differs from the previous estimation, but the main effects remain the same.

[RW]: It is suggested that the figures be shown after they are introduced in the text.

[AU]: We thank the reviewer for this suggestion. Accordingly, throughout the manuscript, we have adjusted the location of figures.

[RW]: In line 362 the explanation of the acronym SP is repeated. Also in line 273. The acronym was presented in line 97.

[AU]:  Thank you for pointing this out. We have removed the redundant explanations of SP.

[RW]: Please correct and unify the format for citing references throughout the article; for example line 85 page 1; lines 384 and 387 in page 11; etc.

         [AU]: We thank the Reviewer for pointing this out. The format has now been corrected and unified.

[RW]: Check the format of all references, for example 25, 38, 40 etc.

         [AU]: : Thank you for pointing this out. The format has been corrected.

  1. Whitham, E. M., Pope, K. J., Fitzgibbon, S. P., Lewis, T., Clark, C. R., Loveless, S., ... & Willoughby, J. O. (2007). Scalp electrical recording during paralysis: quantitative evidence that EEG frequencies above 20 Hz are contaminated by EMG. Clinical neurophysiology, 118(8), 1877-1888.

Reviewer 2 Report

The paper demonstrates that two different cognitive tasks can be associated with the same stimuli, in order to multiplicate the number of possible commands in ERP-based BCI.

The idea that two different paradigms can be combined in a BCI is not novel, and was already suggested for nonclinical application for example [1]. This study is an element of proof that it is indeed possible.

Yet, the main limitation of such a design is probably user fatigue, which is not mentioned in this scenario.

Also, the accuracy that is reported is quite low. In general, we are speaking about 90% of accuracy in a couple of seconds for P300-based BCI [2] - which may me wonder if a simple interface based on 4 stimuli would not have produced better results and less user fatigue. 

Finally, there is no indication that this study follows a randomized design. It could be for example that the difference between the MI and MC potentials is just related to the order of the runs. This couples with the fact that there are only 16 participants. At least, the authors could elaborate on effect size and second-order risk in the discussion.

In brief, there is definitely an element of proof here, but it is not clear how much it is reliable and to what extent it is better than a simple interface with 4 stimuli.

Additionally, since the authors exploit the covariance information of the signal, they should give credit to the work of Barachant and Congedo eg [3, 4]

[1] G. Cattan, A. Andreev, and E. Visinoni, ‘Recommendations for Integrating a P300-Based Brain–Computer Interface in Virtual Reality Environments for Gaming: An Update’, Computers, vol. 9, no. 4, Art. no. 4, Dec. 2020, doi: 10.3390/computers9040092.

[2] M.-C. Corsi, F. Yger, S. Chevallier, and C. Noûs, ‘Riemannian Geometry on Connectivity for Clinical BCI’, in ICASSP 2021 - 2021 IEEE International Conference on Acoustics, Speech and Signal Processing (ICASSP), Jun. 2021, pp. 980–984. doi:

[3] A. Barachant, S. Bonnet, M. Congedo, and C. Jutten, ‘Classification of covariance matrices using a Riemannian-based kernel for BCI applications’, Neurocomputing, vol. 112, pp. 172–178, juillet 2013, doi: 10.1016/j.neucom.2012.12.039.

[4] M. Congedo, L. Korczowski, A. Delorme, and F. Lopes Da Silva, ‘Spatio-temporal common pattern: A companion method for ERP analysis in the time domain’, Journal of Neuroscience Methods, vol. 267, pp. 74–88, 2016, doi: 10.1016/j.jneumeth.2016.04.008.

Author Response

Please find enclosed our revised manuscript, "Mental strategies in a P300-BCI: visuomotor transformation is an option".

We appreciate the time that you dedicated to providing the insightful and highly helpful comments regarding our manuscript. We have substantially changed the manuscript in following the reviewers' suggestions. 

Please, find below our responses to the comments .

RW - Reviewer / AU - authors

[RW]: : The paper demonstrates that two different cognitive tasks can be associated with the same stimuli, in order to multiplicate the number of possible commands in ERP-based BCI.

The idea that two different paradigms can be combined in a BCI is not novel, and was already suggested for nonclinical application for example [1]. This study is an element of proof that it is indeed possible.

[AU]: We thank the Reviewer for the valuable feedback on our manuscript. We hope that the revisions we have made will clarify our manuscript. In particular, we have revised the section where the combination of different paradigms is discussed.

[RW]: Yet, the main limitation of such a design is probably user fatigue, which is not mentioned in this scenario.

[AU]: We thank the Reviewer for the important remark. The experimental sessions consisted of random sequences of 4 runs: 2 MI and 2 MC. The order of these runs was different across subjects. This way we minimized the effect of fatigue on the comparison of different conditions. Yet, we agree that fatigue could have developed within each particular run (that consisted of 90 mental attempts with short rest intervals in between). We have explained in the discussion that this kind of fatigue is a limitation of this design.

[RW]: Also, the accuracy that is reported is quite low. In general, we are speaking about 90% of accuracy in a couple of seconds for P300-based BCI [2] - which may me wonder if a simple interface based on 4 stimuli would not have produced better results and less user fatigue.

            [AU]: We thank the Reviewer for this very important comment. We fully agree with the Reviewer. We ourselves expected to obtain a higher BCI accuracy, so the design should be improved in the future. We have several explanations for this result: 1) our participants were naive in motor imagery and the imagery of button press wasn’t vivid enough to elicit stable cortical modulations; 2) long  experimental runs might have led to fatigue, which could have affected motor-imagery vividness. We have explained these limitations in the discussion. 

One of the main ideas of this article is suggesting the visuomotor transformation BCI paradigm with mental movement attempt performing as a reaction on visual stimulus as potentially useful for neurorehabilitation via stimulation of sensorimotor areas. Here we have shown that the features used for classification were related just with the markers of sensorimotor areas activation.  In further studies we will focus on accuracy improvement.

[RW]: Finally, there is no indication that this study follows a randomized design. It could be for example that the difference between the MI and MC potentials is just related to the order of the runs. This couples with the fact that there are only 16 participants. At least, the authors could elaborate on effect size and second-order risk in the discussion.

            [AU]: We thank the Reviewer for pointing this out. This was not sufficiently explained in the previous methods section. We have revised this section. Specifically, we have explained that all conditions were repeated twice and, importantly, the order of these experimental runs was randomized across subjects. This minimized the effect of the order of runs.

[RW]: In brief, there is definitely an element of proof here, but it is not clear how much it is reliable and to what extent it is better than a simple interface with 4 stimuli.

            [AU]: We fully agree about the need for direct comparison of two BCI-paradigms in one study to compare the speed (bitrate) and accuracy. This will be done in our future work. Yet, we think that the point made in this manuscript -- that our paradigm with two stimuli and four possible outputs reduces the number of the external stimuli and time required to generate an output -- is useful. Additionally, this is not the single point we are making; other results regarding visuomotor transformation in this paradigm are discussed, as well. Overall, our design simplifies the user interface and decreases visual fatigue [1].

  1. G. Cattan, A. Andreev, and E. Visinoni, ‘Recommendations for Integrating a P300-Based Brain–Computer Interface in Virtual Reality Environments for Gaming: An Update’, Computers, vol. 9, no. 4, Art. no. 4, Dec. 2020, doi: 10.3390/computers9040092.

[RW]: Additionally, since the authors exploit the covariance information of the signal, they should give credit to the work of Barachant and Congedo eg [3, 4]

            [AU]:    We thank the Reviewer for the suggestion, which we have implemented. 

Round 2

Reviewer 2 Report

I thank the authors for taking into account my comments. The results and discussion were enhanced and the paper now includes a better description of the experimental design. 

I endorse the publication of this article.